# mTOR Expression in Liver Transplant Candidates with Hepatocellular Carcinoma: Impact on Histological Features and Tumour Recurrence

**DOI:** 10.3390/ijms20020336

**Published:** 2019-01-15

**Authors:** Marta Guerrero, Gustavo Ferrín, Manuel Rodríguez-Perálvarez, Sandra González-Rubio, Marina Sánchez-Frías, Víctor Amado, Juan C. Pozo, Antonio Poyato, Rubén Ciria, María D. Ayllón, Pilar Barrera, José L. Montero, Manuel de la Mata

**Affiliations:** 1Department of Hepatology and Liver Transplantation, CIBERehd, Reina Sofía University Hospital, 14004 Córdoba, Spain; martagmisas1987@hotmail.com (M.G.); victoramadotorres@hotmail.com (V.A.); juanc.pozo.sspa@juntadeandalucia.es (J.C.P.); poyato.antonio@gmail.com (A.P.); pbarrerabaena@gmail.com (P.B.); jlm14005623@gmail.com (J.L.M.); mdelamatagarcia@gmail.com (M.d.l.M.); 2Instituto Maimónides de Investigación Biomédica de Córdoba (IMIBIC), Universidad de Córdoba; 14004 Córdoba, Spain; gusfesa@gmail.com (G.F.); sayuri254@hotmail.com (S.G.-R.); 3Department of Pathology, Reina Sofía University Hospital, 14004 Córdoba, Spain; marinasanchezfrias@gmail.com; 4HPB Surgery and Transplantation, Reina Sofía University Hospital, 14004 Córdoba, Spain; rubenciria@gmail.com (R.C.); lolesat83@hotmail.com (M.D.A.)

**Keywords:** liver transplantation, hepatocellular carcinoma, mTOR, immunosuppression

## Abstract

(1) Background: The mammalian target of rapamycin (mTOR) pathway activation is critical for hepatocellular carcinoma (HCC) progression. We aimed to evaluate the mTOR tissue expression in liver transplant (LT) patients and to analyse its influence on post-LT outcomes. (2) Methods: Prospective study including a cohort of HCC patients who underwent LT (2012–2015). MTOR pathway expression was evaluated in the explanted liver by using the “PathScan Intracellular Signalling Array Kit” (Cell Signalling). Kaplan-Meier and Cox regression analyses were performed to evaluate post-LT HCC recurrence. (3) Results: Forty-nine patients were included (average age 56.4 ± 6, 14.3% females). Phospho-mTOR (Ser2448) was over-expressed in peritumoral tissue as compared with tumoral tissue (ΔSignal 22.2%; *p* < 0.001). The mTOR activators were also increased in peritumoral tissue (phospho-Akt (Thr308) ΔSignal 18.2%, *p* = 0.004; phospho-AMPKa (Thr172) ΔSignal 56.3%, *p* < 0.001), as they were the downstream effectors responsible for cell growth/survival (phospho-p70S6K (Thr389) ΔSignal 33.3%, *p* < 0.001 and phospho-S6RP (Ser235/236) ΔSignal 54.6%, *p* < 0.001). MTOR expression was increased in patients with multinodular HCC (tumoral *p* = 0.01; peritumoral *p* = 0.001). Increased phospho-mTOR in tumoral tissue was associated with higher HCC recurrence rates after LT (23.8% vs. 5.9% at 24 months, *p* = 0.04). (4) Conclusion: mTOR pathway is over-expressed in patients with multinodular HCC and is it associated with increased post-LT tumour recurrence rates.

## 1. Introduction

Hepatocellular carcinoma (HCC) is the most common primary liver malignancy and represents the second cause of cancer-related death worldwide [1]. In early stages, HCC may be treated with a curative intention [2], being liver transplantation (LT) the standard of care for patients with impaired liver function who fulfil Milan criteria (i.e., a single nodule less than 5 cm diameter or up to 3 nodules, less than 3 cm diameter each, without macrovascular invasion or extrahepatic spreading) [3]. However, tumour recurrence after LT occurs in up to 20% of patients and it is associated with dismal prognosis [4,5].

Mammalian target of rapamycin (mTOR) pathway is a key regulator of cellular metabolism and plays an important role in angiogenesis, cell growth and tumour proliferation [6,7]. In HCC, mTOR is the main proliferation pathway in approximately half of patients [8]. Increased expression of mTOR is associated with larger, poorly differentiated and more advanced tumours [9,10]. It has been hypothesized that the upregulation of the mTOR pathway would be associated with increased tumour recurrence rates and shorter survival after potentially curative therapies such as surgical resection or LT [8,10]. In clinical practice, mTOR inhibitors (sirolimus and everolimus) are frequently prescribed in LT patients with HCC as they are potent immunosuppressants, thus effective to prevent organ rejection [11] and could also inhibit the proliferation of remnant HCC cells to prevent tumour recurrence, as shown in in vitro studies [8,12]. However, in clinical practice, the impact of mTOR inhibitors in preventing HCC recurrence after LT is still controversial, with a potential benefit described in observational retrospective studies [13,14,15], which could not be confirmed in prospective studies and randomized trials [16,17]. In addition, mTOR inhibitors are not without risks. A network meta-analysis demonstrated that sirolimus-based immunosuppression regimes are associated with increased mortality rates [18]. Everolimus is thought to be more potent than sirolimus [19] and with an improved safety profile but may still induce hyperlipidaemia and proteinuria in a significant proportion of patients among other adverse events [11,17]. The combination of mTOR inhibitors with sorafenib may be particularly dangerous after LT [20]. An adequate balance between risks and benefits needs to be established in LT patients with HCC regarding the use of mTOR inhibitors. It may well be that only the subset of patients with mTOR pathway over-expression would benefit from immunosuppression protocols based on mTOR inhibitors, thus allowing for a true personalized immunosuppression in LT patients with HCC.

The aims of the present study were: (a) To analyse the mTOR pathway expression in tumoral and peritumoral tissue of HCC patients undergoing LT; (b) To correlate the mTOR pathway expression with histological features of HCC; (c) To explore whether mTOR over-expression modulates the risk of tumour recurrence after LT.

## 2. Results

### 2.1. Population Characteristics and HCC Recurrence

In all, 49 LT patients with HCC were included with a mean age of 56.4 years, being most of them male (*n* = 42, 85.7%). Demographic, clinical characteristics and HCC histological features of the included population are shown in Table 1. Chronic hepatitis C (67.3%) and alcoholic cirrhosis (53.1%) were the main aetiologies of liver disease. Additional aetiologies were chronic hepatitis B (6.1%), autoimmune liver disease (2%), non-alcoholic fatty liver disease (2%), hemochromatosis (2%) and cryptogenic cirrhosis (6.1%). All patients had liver cirrhosis with a pre-transplant MELD score of 13.6 ± 5 points. Forty-five patients (91.8%) had clinical, radiological or endoscopic features of portal hypertension and 34 of them had a previous history of clinical decompensation, being ascites (57.1%) and hepatic encephalopathy (42.9%) the most frequent.

All patients fulfilled Milan criteria based on pre-transplant radiological assessment and 21 patients (42.9%) had a multinodular HCC. Median pre-transplant alpha-fetoprotein was 6.54 ng/dL (IQR 3.4-1922). Bridging locoregional therapies were performed in 31 patients (63.3%) prior to LT. Transarterial chemoembolization (TACE) and radiofrequency ablation (RFA) were the most frequently used modalities (36.7% and 16.3% of patients respectively). In the histological assessment of the explanted liver, 23 patients had multinodular HCC (46.9%). The mean number of nodules was 1.7 ± 0.9 and the diameter of the largest nodule was 3.1 ± 2 cm. Fifteen patients (44.9%) showed moderate/poor histological tumour differentiation. Based on the explanted liver evaluation, 12 patients (24.5%) had a tumour beyond Milan criteria. Microvascular invasion was present in 13 patients (26.5%).

### 2.2. Activation of mTOR Pathway in Tumoral and Peritumoral Tissue and Association with HCC Histological Features

Phospho (p)-mTOR was analysed both tumoral and peritumoral samples together with is activators [p-Akt (Thr308), p-AMPK (Thr172)] and its downstream effectors [p-p70S6K (Thr389) and p-S6RP (Ser235/236)]. Detailed results of the expression of these 7 key proteins in tumoral and peritumoral tissue are shown in Table 2. The whole mTOR cascade expression was increased in peritumoral tissue as compared with tumoral tissue: p-mTOR ΔSignal 22.2% (*p* < 0.001); p-Akt ΔSignal 18.2% (*p* = 0.004); p-AMPK ΔSignal 56.3% (*p* < 0.001); p-P70S6K ΔSignal 33.3% (*p* < 0.001); and p-S6RP ΔSignal 54.6% (*p* < 0.001).

p-mTOR was overexpressed in patients with multinodular HCC, both in tumoral tissue (0.98 ± 0.23 I/mm^2^ vs. 0.8 ± 0.22 I/mm^2^, *p* = 0.01) and in peritumoral tissue (1.19 ± 0.24 I/mm^2^ vs. 0.94 ± 0.23 I/mm^2^, *p* = 0.001). The expression of p-mTOR was not influenced by the diameter of the largest nodule (tumoral *p* = 0.9; peritumoral *p* = 0.37), neither by the total tumour diameter (tumoral *p* = 0.8; peritumoral *p* = 0.38). The presence of microvascular invasion had no significant impact on p-mTOR expression, neither in tumoral tissue (0.96 ± 0.2 I/mm^2^ vs. 0.86 ± 0.3 I/mm^2^, *p* = 0.23), nor in peritumoral tissue (1.03 ± 0.2 I/mm^2^ vs. 1.07 ± 0.3 I/mm^2^, *p* = 0.6). Similarly, histological tumour differentiation did not influence p-mTOR expression (tumour *p* = 0.4 and peritumour *p* = 0.7). Figure 1 summarizes the association between the expression of p-mTOR, p-Akt, p-AMPK, p-p70S6K and p-S6RP and pathological features of HCC. Although there was a trend for increased expression of these proteins in multinodular HCC, poorly differentiated tumours and with microvascular invasion, these differences did not reach statistical significance.

Patients who had received locoregional ablative therapy prior to LT showed reduced expression of p-mTOR within the tumoral tissue (0.81 ± 0.23 I/mm^2^ vs. 1.02 ± 0.2 I/mm^2^, *p* = 0.003) but not in peritumoral tissue (*p* = 0.6). There was a not significant trend for intra-tumour reduced expression of p-Akt (*p* = 0.37), p-AMPK (*p* = 0.2), p-p70S6K (*p* = 0.6) and p-S6RP (*p* = 0.07) in patients with prior locoregional ablation (Figure 1 and Figure 2).

### 2.3. Impact of mTOR Pathway Activation in Tumour Recurrence

Tumour recurrence after LT occurred in 7 patients (14.3%) during follow up. Thresholds of protein expression, both in tumoral and peritumoral tissue, were obtained in order to predict tumour recurrence and they are shown in Table 3. In general, intra-tumour protein expression showed improved accuracy to predict tumour recurrence as compared with peritumour protein expression. P-MTOR, p-AKT and p-AMPK showed the highest AUROC with values ranging from 69.1 to 73.8. The prevalence of intra-tumoral p-mTOR over-expression in the study cohort was 44.9% (22 out of 49 patients). Patients with increased p-mTOR expression in tumoral tissue showed higher tumour recurrence rates during follow-up (23.8% vs. 5.9% at 24 months; log rank *p* = 0.04) (Figure 3). Similarly, increased intra-tumour expression of p-Akt, p-AMPK, p-p70S6K and p-S6RPS was associated with increased risk of tumour recurrence (Figure 3 and Figure 4).

Univariate and multivariate analyses evaluating potential predictors of HCC recurrence after LT are shown in Table 4. In the multivariate Cox’s regression analysis, increased p-AMPK in tumoral tissue [HR 6.9 (95% CI 1.3–37.1), *p* = 0.03] and increased p-S6RP expression in peritumoral tissue [HR 7.5 (95% CI 1.3–43.7), *p* = 0.02] were the only independent predictors of tumour recurrence.

## 3. Discussion

Personalized immunosuppression in HCC patients undergoing LT would allow to avoid graft rejection while reducing tumour recurrence rates but tissue biomarkers remain elusive and randomized trials focused in high-risk patients are lacking. In the present study, the mTOR pathway was upregulated in the explanted liver of a significant proportion of patients with HCC undergoing LT and this finding was associated with increased tumour recurrence rates. Histological tumour differentiation and microvascular invasion were not linked with p-mTOR overexpression and therefore these histological features per se should not motivate the prescription of mTOR inhibitor-based immunosuppression in clinical practice.

LT is the best therapeutic options for selected patients with early stage HCC and underlying liver cirrhosis, as it allows for survival rates of 70–80% at 5 years, which are far superior to other therapeutic modalities [21]. Tumour recurrence may still occur in up to 20% of patients and may have a negative prognostic impact. A refined selection and prioritization of candidates for LT, as well as indication of ablative locoregional therapies, may decrease post-LT HCC recurrence rates but they should be accompanied by a personalized immunosuppression regimen [22]. To date, the only immunosuppression strategy with a potential effect in reducing the risk of HCC recurrence is the early minimization of calcineurin inhibitors [23,24].

MTOR is a serine/threonine protein kinase, which belongs to the phosphoinositide 3-kinase (PI3K) family [25] and its activation may be triggered by growth factors, cytokines, toll-like receptors, hypoxia and DNA damage. The mTOR pathway has been involved in cell growth, aging and metabolism. It comprises the mTOR complex 1 (mTORC1) and the mTOR complex 2 (mTORC2), being the former responsible for cell growth and proliferation, while the latter is associated with cell survival and metabolism [26,27]. The mTOR pathway is aberrantly upregulated in HCC [8,9,10]. In the largest study published hitherto including 351 HCC histological samples from resected specimens or explanted livers, Villanueva and cols. estimated that mTOR pathway would be upregulated in approximately 50% of cases [8] aligning with the present study, in which 22 patients (44.9%) had increased p-mTOR expression.

In recent years, mTOR inhibitors (sirolimus and everolimus) have been approved as maintenance immunosuppression agents after LT. In combination with tacrolimus minimization, mTOR inhibitors proved to be safe and effective as renal sparing agents [11]. Since mTOR inhibitors behave as antiproliferative agents in experimental models, they were hypothesized as ideal immunosuppressive drugs for LT patients with HCC [28]. However, the only randomized trial evaluating mTOR inhibitors in this clinical context (i.e., the SILVER trial [16]), which was sufficiently powered and with a prolonged follow-up, did not show a benefit in terms of recurrence-free survival rates. However, the subset of patients within Milan criteria experienced a delay in tumour recurrence, thus suggesting that the benefit of mTOR inhibitors may be restricted to a subgroup of patients still to be defined [16]. The present study reinforces this hypothesis, as the mTOR pathway was found heterogeneously expressed in LT candidates with HCC. The antiproliferative effect of mTOR inhibitors may be underestimated if these drugs are used in an unselected cohort of HCC patients. Although mTOR expression was increased in patients with multinodular disease, aligning with previous reports [8,10], it was not associated with other established histological features indicating worse prognosis such as microvascular invasion or tumour differentiation, probably because of a confounding effect of pre-LT locoregional ablative therapies, which are more frequently used nowadays than in previous series [8,10]. In absence of high-quality evidence, the rationale of prescribing mTOR inhibitors in all patients may not be justified, neither it is to restrict their use to patients with histological features of worse prognosis. We strongly believe that mTOR inhibitors should be preferably indicated in patients with proven histological mTOR pathway upregulation within the tumour but a randomized trial focused in this subpopulation is needed to reinforce this rationale.

In the present study, the expression of p-mTOR, together with its activators (i.e., p-Akt, p-AMPK) and downstream effectors (i.e., p-p70S6K and p-RPS6) was more intense in the tumour edge. Although challenged by some previous reports [9,29], mTOR overexpression in the tumour edge would be expected since tumour expansion is classically considered centrifugal and epithelial to mesenchymal transition occurs primarily in the tumour edge [30]. In addition, there may be an intense inflammatory reaction in the peritumour area driven by immune cells striving to counteract tumour spreading [31,32] and which proliferation may be at least partially dependent from the mTOR pathway. In other words, the mTOR activity within the tumour would mirror the actual tumour proliferation status while the mTOR expression in the tumour edge would be confounded by other coexisting phenomena such as peritumour innate inflammation response. This may explain why, in the present study, the mTOR pathway expression within the tumour and not in the peritumour, was associated with HCC recurrence after LT. In future randomized trials, the mTOR pathway expression should be assessed within the tumour in order to avoid the potential confounding effect of peritumour inflammation.

Locoregional ablative therapies are recommended in patients with HCC included in the waiting list provided that the tumour diameter is larger than 2 cm or if the expected length within the waiting list is longer than 6 months [3]. In clinical practice, most of patients would fall within these criteria and therefore locoregional ablative therapies would be used in most cases unless technically unfeasible (i.e., severely impaired liver function or anatomic caveats) in order to avoid tumour progression and derived dropout from the waiting list. In the present study, about 60% of patients underwent locoregional ablation prior to LT, mainly trans-arterial chemoembolization and they showed reduced p-MTOR expression within the tumour. It may well be that central tumour necrosis and subsequent mTOR inhibition would be responsible for the reduced tumour recurrence rates found in patients who received bridging locoregional ablative therapies [33]. Future studies may explore the effect of mTOR tissue expression after tumour ablation and its relationship with tumour recurrence.

The main limitation of this study is the reduced sample size, which may explain why microvascular invasion and tumour differentiation did not reach statistical significance as predictors of tumour recurrence. In addition, as an observational study, no intervention was differentially applied in patients with and without mTOR upregulation. All patients received everolimus at low dose (i.e., with trough concentrations close to the lower target range of 3 ng/mL) during follow-up as part of a multicentre prospective observational study [17]. Whether higher dose of everolimus in patients with mTOR upregulation would have resulted in improved outcomes remain to be determined.

## 4. Materials and Methods

### 4.1. Patients

A prospective consecutive cohort of adult patients with HCC who underwent LT from October 2012 to December 2015 was included. Exclusion criteria were as follows: HIV positive, combined organ transplantation, re-transplantation and perioperative death (i.e., within the first month after LT). Only patients within Milan criteria in the pre-transplant imaging assessment were accepted as candidates for LT and downstaging strategies were allowed in carefully selected patients with borderline tumour burden and favourable response to locoregional therapies. Locoregional ablative therapies were indicated to prevent dropout in patients with multinodular disease or with a single nodule larger than 20 mm unless technically unfeasible. All patients were required to sign an informed consent document in order to enter the study. This project was conducted according to Declaration of Helsinki and was approved by the Andalusian ethics committee on 30 March 2015 as part of a broader research initiative (PI11/02867and PI14/01469).

Patients were followed up until June 2018 with a median follow-up was 39 months. All patients received tacrolimus-based immunosuppression, with trough concentrations aimed at 6–10 ng/mL within the first month after LT and 4–8 ng/mL thereafter, as higher doses may increase the risk of HCC recurrence [23]. Mycophenolate mofetil was initially associated at 500 mg bid and switched to everolimus between post-operative days 15 to 21 as part of an observational study [17]. Steroids were progressively tapered until completely withdrawn within the first 6 months post-transplant. Screening of HCC recurrence consisted in abdominal ultrasound and serum alpha-fetoprotein every 3 months within the first year and every 6 months thereafter. Any suspicious liver nodule detected in the ultrasound or rising alpha-fetoprotein otherwise unexplained motivated the indication of liver magnetic resonance or computed tomography. Imaging and histological criteria to diagnose HCC were based on current international guidelines [3].

### 4.2. Histopathology

Histological evaluation of the explanted liver was systematically performed by an experienced pathologist to determine the number of HCC nodules, the diameter of the largest nodule, total tumour burden (sum of diameter of each recorded nodule), histological differentiation according to the Edmonson scale [34], macrovascular invasion and microvascular invasion (defined either as a tumour emboli within a peritumoral vessel or as a satellite nodule surrounded by endothelium -positive for CD34 staining [35]).

### 4.3. Protein Extraction from FFPE Tissues

We obtained peritumour (cirrhotic non-tumoral tissue) and tumour tissue samples from all patients (98 samples). Protein samples were extracted from formalin-fixed paraffin-embedded (FFPE) liver tissue by the Qproteome^®^ FFPE Tissue Kit (Qiagen, Hilden, Germany), which ensures optimized extraction efficiency and enables extraction of full-length, intact proteins from FFPE blocks for proteomic studies. This protocol includes the deparaffinization and hydration of FFPE tissue sections and subsequent incubation of tissue samples in a lysis buffer at 2 different temperatures in order to reverse formalin cross-linking and untangle protein molecules. In our experience, deparaffinization of FFPE tissue sections directly cut from an FFPE sample block did not obtain optimal results, probably due to the limited size of the sample and the increased number of sections needed (up to 3 sections, each with a thickness of 10 µm and an area of up to 100 mm^2^). In order to overcome this caveat, we used FFPE tissue sections mounted on microscope slides, which had been previously incubated at 60 °C for 1 h and then immersed into suitable reservoirs containing xylene (3 × 10 min). This approach ensured a correct deparaffinization of the tissue sections. Next, we hydrated them by transferring the slides to reservoirs containing 100% ethanol (2 × 10 min), 96% ethanol (2 × 10 min), 70% ethanol (2 × 10 min), double-distilled water (2 × 5 min) and PBS (2 × 10 min) (PBS tablets pH 7.2, Panreac, Barcelona, Spain). To extract the protein samples, we took the slides from the PBS-containing reservoir and placed it into a sterile microcentrifuge tube containing 100 µL extraction buffer (supplemented with β-mercaptoethanol). After incubating on ice for 5 min, samples were included on a heating block, firstly at 100 °C for 20 min and then at 80 °C for 2 h with agitation at 750 rpm. Finally, samples were centrifuged at 14,000× *g* for 15 min at 4 °C and the resulting supernatants, which contained the extracted proteins, were transferred to new tubes. Protein yield was quantified by the Bradford method after appropriate sample dilution.

### 4.4. Detection and Quantification of Protein Related to the mTOR Signalling Pathway

The mTOR pathway expression was analysed by using the PathScan^®^ Intracellular Signalling Array Kit (Cell Signalling, Danvers, MA, USA) (Figure 5). This is a slide-based antibody array founded upon the sandwich immunoassay principle. The array allows for the simultaneous detection (in duplicate) of 18 important and well-characterized signalling molecules when phosphorylated or cleaved, which have been related to the mTOR signalling pathway and involved the following cascades: MAPK/Erk [p-ERK1/2 (Thr202/Tyr204), p-p38 (Thr180/Tyr182), p-SAPK/JNK (Thr183/Tyr185)], Akt [p-Akt (Thr308 and Ser473), p-AMPK (Thr172), p-PRAS40 (Thr246), p-Bad (Ser112), p-GSK-3 (Ser9)], Stat [p-Stat1 (Tyr701), p-Stat3 (Tyr705)], p38 [p-p38 (Thr180/Tyr182), p-HSP27 (Ser78)] and p53 [p-p53 (Ser15), p-caspase-3 (Asp175), p-PARP (Asp214)]. The array also includes p-mTOR (Ser2448) and downstream molecules that reflect the mTOR pathway activation, such as p-p70S6K (Thr389) and p-S6RP (Ser235/236). In addition, each array contains positive and negative controls by triplicate. In order to make results more comprehensive and simple, we selected 7 key representative proteins of the pathway including p-MTOR, 3 of its activators and 3 of its downstream effectors, which had the best predictive capacity of tumour recurrence.

For protein detection, we first diluted protein samples extracted from FFPE liver tissues at 0.3 µg/µL with diluent buffer. After blocking for 30 min at room temperature, we added 75 µL of protein sample (22.5 µg; best protein dilution/quantity we tested) to each well/array and we incubated overnight at 4 °C on an orbital shaker. Then, we consecutively incubated with detection antibody cocktail for 1 h and HRP-linked streptavidin for 30 min at room temperature on an orbital shaker. We washed for 5 min (×4) with array wash buffer after each step. Then, the wash buffer was decanted and the slide covered with LumiGLO^®^/Peroxide reagent and a sheet protector. Images of the slide were captured immediately thereafter by using the digital imaging system LAS4000 (FujiFilm, Tokio, Japan) for detecting chemiluminescent signal. The best slide image was analysed and the spot intensities were quantified by using the Quantity One image analysis software (Bio-rad, Berkeley, CA, USA). We used negative controls to define the background and positive controls for the relative quantification of protein expression.

The investigator responsible for tissue processing and analysis was blinded to the clinical information of the patients during the whole process.

### 4.5. Statistical Analysis

Variables were displayed in frequency tables or expressed as mean and standard deviation, except in those variables with asymmetric distribution in which median and interquartile range were used. Appropriate contrast tests were applied according to the characteristics of the involved variables: chi square test for frequencies, Student’s t or ANOVA tests for quantitative variables and Mann-Whitney’s U or Kruskal-Wallis for asymmetric distributions. The area under the receiver operating characteristics (ROC) curve (AUC) analysis was used to define optimal cut-off for protein expression to predict tumour recurrence. The optimal threshold was selected by using the Youden Index. Kaplan-Meier curves (Log rank) and Multivariate Cox regression analyses were used to determine predictors of tumour recurrence, while controlling for potential confounders. Those variables with *p* ≤ 0.25 in the univariate analysis entered the initial multivariate model. Covariates were removed from the model in a backward stepwise process. All possible interactions among covariates were tested. Variables with a *p* value between 0.05 and 0.20 were screened to identify potential confounding factors and further removed from the model if they did not behave as such. Every hypothesis tested was two tailed and considered significant if *p* < 0.05. Statistical analysis was performed using SPSS 20.0 (IBM, Chicago, IL, USA).

## 5. Conclusions

In conclusion, the mTOR pathway is upregulated in a subset of patients with HCC undergoing LT, particularly within the tumour edge where cellular growth and proliferation are more intense. MTOR histological expression is increased in patients with multinodular disease but it is not influenced by other histological features classically associated with worse prognosis such as microvascular invasion or poor tumour differentiation. Finally, mTOR upregulation within the tumour is associated with increased tumour recurrence rates. In light of these data, mTOR inhibitor-based immunosuppression could be considered in patients with mTOR over-expression in whom adverse effects of these drugs may be overcome by a potential benefit of decreased tumour recurrence rates. A randomized trial focused on patients with mTOR over-expression within the tumour is required to strengthen the evidence and to advance towards a true personalized immunosuppression in HCC.

## Figures and Tables

**Figure 1 ijms-20-00336-f001:**
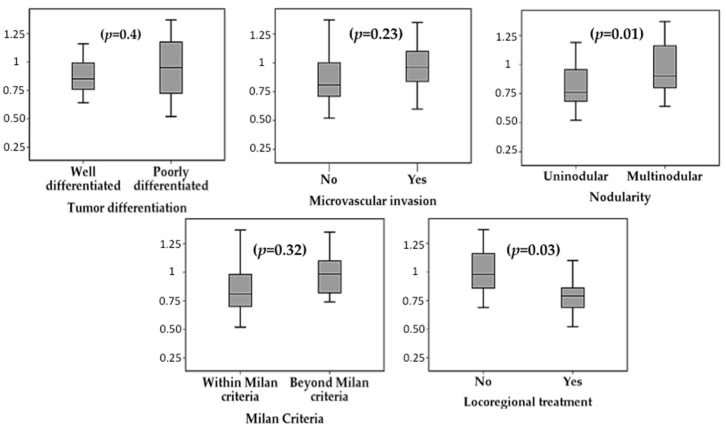
Phospho-mTOR expression in tumoral tissue (I/mm^2^) according to histological features of hepatocellular carcinoma as determined in the explanted liver: histological tumour differentiation, microvascular invasion, tumour nodularity, fulfilment of Milan criteria and prior locoregional ablative therapy.

**Figure 2 ijms-20-00336-f002:**
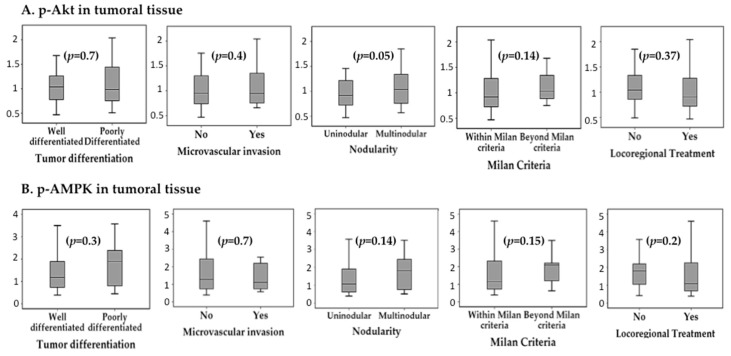
Association between histological features of hepatocellular carcinoma and average intra-tumour expression (I/mm^2^) of the evaluated mTOR pathway components: (**A**) p-Akt, (**B**) p-AMPK, (**C**) p-70S6K and (**D**) p-6SRP.

**Figure 3 ijms-20-00336-f003:**
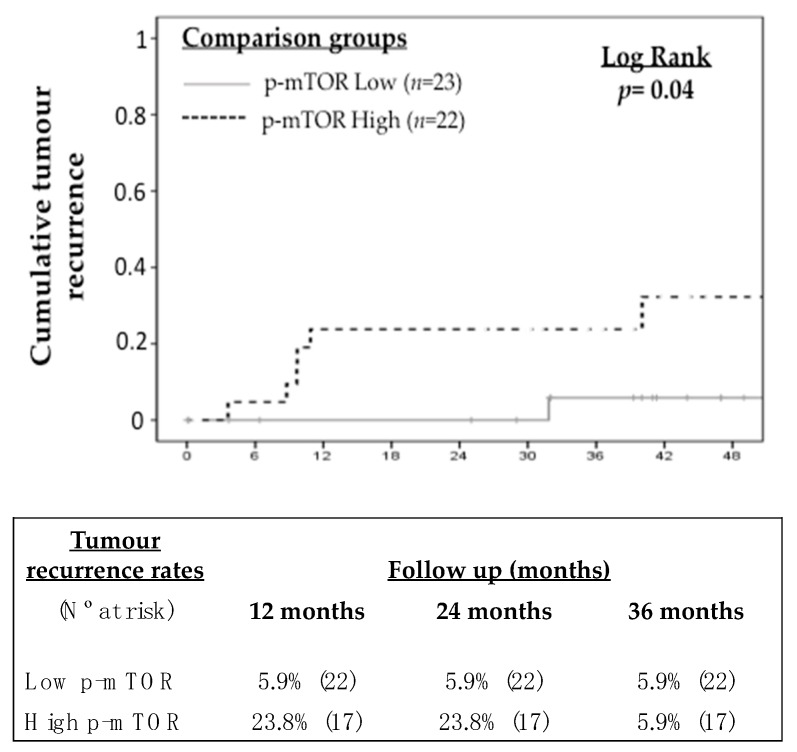
Kaplan-Meier curve showing the impact of intra-tumour p-mTOR expression on the recurrence of hepatocellular carcinoma after liver transplantation.

**Figure 4 ijms-20-00336-f004:**
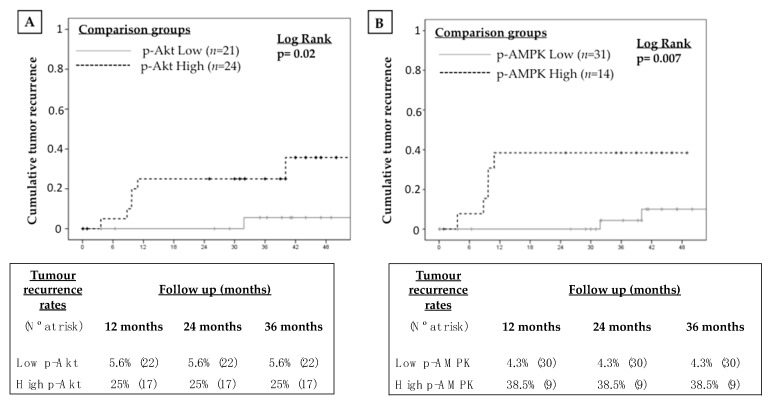
Kaplan-Meier curves showing the impact of the over-expression of selected mTOR pathway components within the tumour tissue on tumour recurrence after liver transplantation: (**A**) p-Akt, (**B**) p-AMPK, (**C**) p-70S6K and (**D**) p-S6RP.

**Figure 5 ijms-20-00336-f005:**
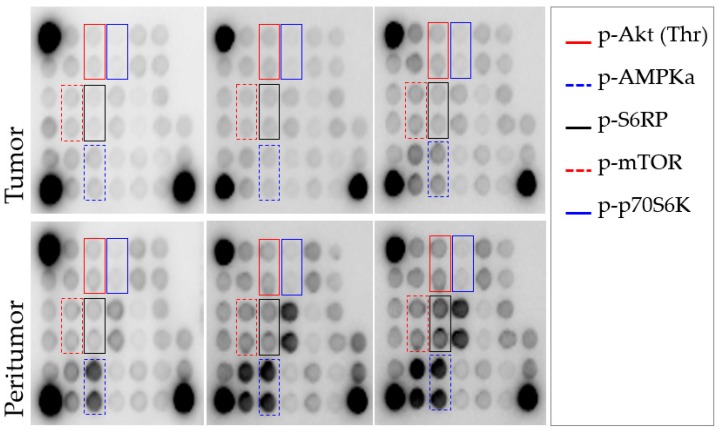
Representative panels showing the output from the PathScan intracellular signalling array in 6 tissue samples: 3 tumoral samples (**upper panel**) and 3 peritumoral samples (**lower panel**). The whole array included immobilized antibodies against 18 proteins when phosphorylated or cleaved (by duplicate), which are related to the mTOR-signalling pathway. Each protein is represented by a pair of spots. Those proteins with a statistically different intensity between tumoral and peritumoral samples are marked with colour rectangles in both arrays. Abbreviations: p-Akt, phospho protein kinase B or PKB; p-AMPKa, phospho protein kinase AMP-activated catalytic subunit alpha 1 or PRKAA1; p-S6RP, phospho S6 Ribosomal Protein; p-mTOR, phospho mechanistic target of rapamycin; p-p70S6K, phospho ribosomal protein S6 kinase beta-1 or S6K1.

**Table 1 ijms-20-00336-t001:** Demographic, clinical characteristics and histological features of 49 patients with hepatocellular carcinoma who underwent liver transplantation.

**Baseline Characteristics (*N* = 49)**	
**Age, mean (SD)**	56.4 (7.2)
**Gender**	
**Male, *n* (%)/Female, *n* (%)**	42 (85.7)/7 (14.3)
**Aetiology of liver disease**	
**HCV, *n*** **(%)**	33 (67.3)
**Alcohol, *n*** **(%)**	26 (53.1)
**HBV, *n*** **(%)**	3 (6.1)
**Autoimmune liver disease, *n*** **(%)**	1 (2)
**NAFLD, *n*** **(%)**	1 (2)
**Hemochromatosis, *n*** **(%)**	1 (2)
**Cryptogenetic, *n*** **(%)**	3 (6.1)
**MELD score, mean (SD)**	13.6 (5.1)
**Previous hepatic decompensation, *n* (%)**	34 (69.4)
**Ascites, *n*** **(%)**	28 (57.1)
**Hepatic encephalopathy, *n*** **(%)**	21 (42.9)
**Variceal bleeding, *n*** **(%)**	11 (22.4)
**SBP, *n*** **(%)**	7 (14.3)
**HRS, *n*** **(%)**	3 (6.1)
**Bridging LRT, *n* (%)**	31 (63.3)
**TACE, *n* (%)**	15 (36.7)
**RFA, *n* (%)**	8 (16.3)
**TACE + RFA, *n* (%)**	6 (12.2)
**Liver resection, *n* (%)**	2 (4.1)
**HCC radiological features**	
**Uninodular/multinodular, *n* (%)**	21 (42.9)/21 (42.9)
**Number of nodules, mean (SD)**	1.62 (0.7)
**DLN, mean (SD)**	2.5 (0.9)
**TTB, mean (SD)**	3.3 (1.5)
**HCC histological features**	
**Uninodular/multinodular, *n* (%)**	26 (53.1)/23 (46.9)
**Number of nodules, mean (SD)**	1.7 (0.9)
**DLN, mean (SD)**	3.1 (2)
**TTB, mean (SD)**	4.1 (2.5)
**Well differentiated or complete necrosis, *n* (%)**	22 (44.9)
**Beyond Milan criteria, *n* (%)**	12 (24.5)
**Microvascular invasion, *n* (%)**	13 (26.5)
**Satellitosis, *n* (%)**	2 (4.1)
**Macrovascular invasion, *n* (%)**	2 (4.1)

Abbreviations: HCV: hepatitis C virus; HBV: hepatitis B virus; NAFLD: non-alcoholic fatty liver disease; SBP: spontaneous bacterial peritonitis; HRS: hepatorenal syndrome; LRT: Locoregional treatment; TACE: Transarterial Chemoembolization; RFA: Radiofrequency ablation, DLN: Diameter of the largest nodule; TTB: total tumour burden.

**Table 2 ijms-20-00336-t002:** Protein expression in tumoral and peritumoral tissue (I/mm^2^). Abbreviations: SD: standard deviation.

Protein	Location	Mean (SD)	ΔSignal (%)	*p*
p-mTOR	Tumoral	0.9(0.2)	22.2	<0.001
	Peritumoral	1.1 (0.3)		
p-AKT	Tumoral	1.1 (0.4)	18.2	0.004
	Peritumoral	1.3 (0.5)		
p-AMPK	Tumoral	1.6 (1.1)	56.3	<0.001
	Peritumoral	2.5 (1.7)		
p-P70S6K	Tumoral	0.3 (0.14)	33.3	<0.001
	Peritumoral	0.4 (0.2)		
p-S6RP	Tumoral	1.1 (0.5)	54.6	<0.001
	Peritumoral	1.7 (1)		

**Table 3 ijms-20-00336-t003:** Optimal threshold, sensitivity, specificity and area under ROC curve (AUC).

Protein	Origin	Threshold	Sensitivity (%)	Specificity (%)	AUC
p-mTOR	Tumoral	0.85	85.70	57.90	71.80
	Peritumoral	1.18	42.90	73.70	58.27
p-AKT	Tumoral	1.00	85.70	57.90	69.17
	Peritumoral	1.96	28.57	84.20	53.57
p-AMPK	Tumoral	2.19	71.40	76.30	73.87
	Peritumoral	0.70	100.00	13.2	56.58
p-P70S6K	Tumoral	0.47	42.90	89.50	66.17
	Peritumoral	0.41	71.40	57.90	63.35
p-S6RP	Tumoral	1.05	71.40	63.20	67.29
	Peritumoral	3.52	28.60	97.40	62.97

**Table 4 ijms-20-00336-t004:** Univariate and multivariate Cox’s regression analyses. Variables included in the final multivariate model are highlighted in bold.

Variable	Univariate Analysis			Multivariate Analysis		
	HR	95% CI	*p*	HR	95% CI	*p*
LRT	0.76	0.17–3.4	0.72	-	-	-
Multinodular	1.55	0.35–6.9	0.6	0.42	0.06–2.8	0.37
DLN	0.86	0.54–1.38	0.55	-	-	-
TTB	1.05	0.79–1.39	0.74	-	-	-
Beyond Milan	1.11	0.22–5.74	0.90	-	-	-
Microvascular invasion	2.37	0.53–10.61	0.26	2.18	0.4–11.9	0.37
Poorly differentiated	1.94	0.38–10.02	0.43	-	-	-
High p-Akt Tumoral	6.27	0.75–52.4	0.09	1.64	0.11–24.9	0.72
High p-Akt Peritumoral	1.49	0.33–6.74	0.6	-	-	-
High p-AMPK Tumoral	7.05	1.36–36.64	0.02	**6.9**	**1.3–37.1**	**0.03**
High p-AMPKa Peritumoral	25.33	0.002–292543	0.5	-	-	-
High p-mTOR Tumoral	6.58	0.79–54.72	0.08	2.08	0.21–20.9	0.53
High p-mTOR Peritumoral	2.04	0.5–9.13	0.4	-	-	-
High p-P70S6K Tumoral	5.49	1.2–24.9	0.03	1.19	0.07–19.41	0.9
High p-P70S6K Peritumoral	3.29	0.63–17.18	0.16	2.25	0.34–14.81	0.4
High p-S6RP Tumoral	3.57	0.7–18.41	0.13	0.84	0.1–7.28	0.87
High p-S6RP Peritumoral	8.01	1.52–41.88	0.01	**7.5**	**1.3–43.7**	**0.02**

Abbreviations: LRT: Locoregional therapy; DLN: Diameter of the largest nodule; TTB: total tumour burden.

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
