# Peer review of "mTOR Expression in Liver Transplant Candidates with Hepatocellular Carcinoma: Impact on Histological Features and Tumour Recurrence"

_ijms, 2019, doi:10.3390/ijms20020336_

Round 1
Reviewer 1 Report
The study sounds interesting and is well written.
One main point should be stressed (page 2 Line 62):
Everolimus has been found to be futile as second line therapy of HCC.
Moreover, different single center studies suggested and a meta-analysis confirmed that there is justified concern about side effects (in particular bleeding) when m-Tor inhibitors are added to sorafenib as treatment of HCC recurrence after LT. This is another setting in which the results of the present study could have importance.
Page 11, Line 186: I would absolutely be more dubitative about trhe sentence regarding Ref 20.
Author Response
Please, see the attached pdf document.

Reviewer 2 Report
The study was performed in order to evaluate the expression of proteins, participated in activation of mTOR pathway and to analyze the relationship of survival and tumor recurrence in patients after the liver transplantation (LT). Forty nine patients (42 males and 7 females) were subjected to the analysis. Unexpectedly, authors performed not the immunohistochemical evaluation, but use the PathScan Intercellular Signaling Array kit (antibody array), and the data are presented as signal intensity/mm2.
Authors detected that mTOR pathway is activated in the peritumoral regions of the LT patients and the expression of components of mTOR pathway is associated with increased recurrence in LT patients. mTOR pathway components were positive in about half of patients. Authors suggested that patients with activated mTOR should be sensitive to the therapy with mTOR inhibitors.
Major Comments
1. In the present study, the number of patients with HCCs subjected to analysis was not high (49 patients). Furthermore, I did not find the controls. For example, liver samples of patients with metastasis from gastrointestinal tumors are usually used for the comparison. Authors compared expression of mTOR pathway-involved proteins in tumoral and peritumoral areas. Furthermore, peritumoral area appeared to have high levels of cell proliferation, as mentioned by the authors. Why non-tumoral areas were not analyzed? At least some areas must be present in FFPE blocks.
2. In the Table 1 the data of 42 male subjects are presented. However, I did not find any data of females. Where did they go?
3. As I understood, 45 patients were subjected to survival analysis as well as comparison of tumor recurrences in low and high target protein expression groups. Where did the remained 4 patients disappear?
4. Authors explain increase of P-mTOR and other phosphorylated proteins in the peritumoral areas by the high levels of cell proliferation and inflammation. Does this mean that the increased recurrence rates are dependent on the mTOR activation outside the tumor??? Furthermore, no positive relationship was found between mTOR activation and tumor differentiation as well as micrometastasis in the liver, what is really strange, because in many studies positive relationship has been reported between tumor differentiation and recurrence (poor>well), and presence or absence of micrometastasis and recurrence (presence>absence). This discrepancy of the data should be addressed.
5. In the analyses, HCV, HBV-positive, alcohol-drinkers and NAFLD patients are combined in the same cohort, what is quite strange. The etiology of cancer is too different to combine all patients.
6. The results of the PathScan Intercellular Signaling Array, for instance Mean and SD values of Intensity/mm2 (Table 2), are quite questionable. The differences of P-mTOR, P-Akt, P-P70S6K expressions in tumoral and peritumoral areas are very small. Authors mentioned, that expression of mTOR-related proteins was observed in about half of the patients with HCC. How the statistical analyses were performed? With all patients cohort, or only half of it? Evaluation and proof of the obtained results is strongly recommended. For example, checking by the immunohistochemical analyses. The presentation of the immunohistochemical data as visible proof is highly recommended.
7. Explanation of Statistical analysis of the data is very non-informative. Why there are no controls? By what method peritumoral and tumoral areas were compared? There is no clear explanation. The number of samples applied in the present study is not high, and the application of appropriate Statistical analysis is of high importance.
8. In many places, it is impossible to understand, whether authors describe the P-mTOR, P-AMPK, P-Akt, P-P70 and P-S6 or the mTOR, AMPK, Akt, p70 and S6 (Tables 2, 3, 4, Figure 3 and in the text). It must be clearly stated inside the figures and Tables (not only in the legends) and in the manuscripts text, which proteins were analyzed (phosphorylated or not). Otherwise, readers of the paper would be confused.
9. It is very strange, that authors have use P-Akt(Thr308), but not the P-Akt(Ser473), which is reportedly involved in mTOR pathway, but P-Akt(Thr308) is needed to be proved still. This needed to be explained at least.
10. In the Materials and Methods section explanations are not complete. More detailed explanation is needed. For instance, there is no clear information on antibodies cocktails. There must be precise information given.
11. The pictures are of low quality. In my version, pictures (especially in figures 1, 2 and 3) are very small and letters are almost impossible to read. I strongly recommend to revise them.
12. Figure 5 is not informative. What is the difference between upper and lower panels? And what is the difference between the upper and lower three panels. What proteins, besides mTOR–related are presented in the panels? Is this the only two samples of tumoral and peritumoral areas of one patient? Picture is very difficult to understand.
13. The improvement of English of the paper by the native speaker is needed.
Author Response
Please see the pdf document attached.

Round 2
Reviewer 2 Report
Authors have corrected the manuscript accordingly to the suggestions and it can be now accepted for publication.
English style of the paper has been improved, but still there are small misstypes to be corrected. Please check all spaces between words, words and numbers, words and brackets, etc. In the Figure 2, "0" are omitted in the graphs, what is better to correct making the same as in other Figures.
Please use the same word ("tumor" or "tumour") all through the manuscript.
Author Response
Dear Reviewer:
We are glad that you found most of your initial concerns appropriately addressed in the previous version of the manuscript. We have carefully revised the paper and corrected some typos. The word "tumour" has been replaced by "tumor" along the manuscript. In addition, the figure 2 has been modified as you requested.
We hope you will find our manuscript acceptable for publicacion in its present version.
Best regards,
Manuel Rodríguez-Perálvarez, MD, PhD
On behalf of all co-authors